# Social Support and Depression among Stroke Patients: A Topical Review

**DOI:** 10.3390/ijerph20247157

**Published:** 2023-12-08

**Authors:** Haoyu Zhou, Erin R. Kulick

**Affiliations:** Department of Epidemiology and Biostatistics, Temple University College of Public Health, Philadelphia, PA 19122, USA; haoyu.zhou@temple.edu

**Keywords:** social support, depression, stroke, stroke patients

## Abstract

Research has shown a protective association between social support and depression, depression among stroke patients, and health impacts of depression. Despite this, not much is known about the effect of social support on depression among stroke patients. This review aims to summarize the current research examining the association between social support and depression among stroke patients. A literature search was performed in PubMed to find original peer-reviewed journal articles from 2016 to 12 March 2023 that examined the association between social support and depression among stroke patients. The search terms were depression and “social support” and stroke, which lead to 172 articles. After abstract review, seven observational studies that studied the target association among stroke patients were selected. One additional study was found using PsycINFO as a complementary source with the same search strategy and criteria. Overall, a negative association was found between social support and depression among stroke patients in eight studies, with more social support leading to lower rates of depression post-stroke. The other study did not find a statistically significant association. Overall, the results of recent studies suggest that social support is negatively associated with depression among stroke patients. In most studies, this association was statistically significant. The findings suggest the importance of improving social support perceived by stroke patients in the prevention of depression after the occurrence of stroke.

## 1. Introduction

Stroke is an abrupt neurological outburst following impaired blood perfusion to the brain or bleeding due to rupture of a blood vessel [1,2]. The impaired blood perfusion causes the central nervous system to suffer from an acute focal injury, which can then lead to a neurological deficit [3]; in the bleeding case, a hematoma is formed that causes compression to the brain tissue and increase of the intracranial pressure [2,3]. Stroke is the second leading cause of death worldwide as of 2019 [4]. The global mortality rate of stroke is about 5.5 million, and about half of stroke survivors suffer from some level of long-term disability [4]. In the United States (USA), stroke remains the 5th cause of death [5]. A stroke occurs every 40 s, and death from stroke every 4 min in the USA [6]. Based on the mechanisms, stroke typically consists of three types—ischemic stroke, hemorrhagic stroke, and transient ischemic attack (TIA) [1]. Post-stroke complications are multiple—post-stroke seizures, partial paralysis, bladder dysfunction, bowl dysfunction, and sleep disturbance may start immediately after stroke onset and can last for years [1,7]. Apart from the physical complications, psychosocial complications occur as well. Depression is one of the major mental problems following stroke [1,7,8,9]. Depression is a mental disorder that leads to constant sad feelings and loss of interest [10]. In addition to sad feelings and loss of interest, people with depression may typically experience somatic symptoms, including changes in appetite, sleeping disorder, chronic pain in joints, the back and limbs, fatigue, and gastrointestinal problems [11,12]. Changes in appetite and gastrointestinal problems may then result in other health problems such as diabetes [13]. Besides, depressive symptoms are associated with increased social isolation [14]. Even worse, depressed patients may think about and attempt suicidal behavior [15]. The prevalence of post-stroke depression (PSD) occurs in 18–31% of stroke patients, according to results of several meta-analysis studies [16,17,18]. Worldwide, depression affects approximately 3.8% of the total population, and around 5.0% among adults (18 years or older) [19]. Additionally, PSD is highly associated with post-stroke physical complications [1,7]. For example, bladder dysfunction leads to urine incontinence, which is highly associated with an increased risk of depression [20,21]. Even more concerning, depression has been found associated with decreased adherence to medication treatment for stroke, leading to increased risks of stroke recurrence and mortality [22,23,24].

As introduced above, depression does not just impact mental health, but also introduces somatic symptoms and problems in daily life [11,12,13,14,15,25,26]. However, what makes depression worse is that 69% of patients who met the criteria of depression reported only somatic symptoms when they paid a visit to health professionals, according to a study by Simon et al. [27]. Patients reporting only somatic symptoms instead of psychological symptoms such as depressed mood can lead to undiagnosed depression [27,28]. This, among other factors, including concerns of being recommended antidepressants and being referred to a psychiatrist, often hinders early diagnosis and treatment of depression [27,28,29,30,31]. The underreporting is a particularly serious problem among stroke survivors, whose depression is often co-morbid with physical ailments and associated with the recurrence and mortality [22,23,24]. The underdiagnosis also indicates that depression is in fact more prevalent among the populations than indicated by the observed numbers, including among stroke patients [32]. It is essential to examine the modifiable factors associated with depression for the purpose of prevention and treatment.

The presence of high social support is thought to be particularly helpful in PSD. Social support is a psychological construct that refers to the provision of assistance or comfort that are provided and are accessible to individuals, typically thought to help individuals deal with biological and psycho-social stressors [33]. Social support has multiple operationalized meanings, which can include structural components such as social integration, which means being a part of and participating in social networks [34,35]. Social support can also refer to functional components, such as types of interactions between individuals [36]. In most studies, social support is conceptualized as perceived social support, which is typically measured by psychological scales that cover the components above [37,38,39,40].

By affecting coping strategies, meeting daily social needs, and encouraging community participation, social support has been found to be an important factor for both the acute recovery and daily life of stroke patients [41,42,43,44,45]. Despite this, little is known about the association between social support and depression/depressiveness among stroke patients, either at the acute phase or in a longer period. Multiple studies have evaluated the association between social support and mental health among various populations, such as the elderly or the general population [46,47,48,49,50,51]. Findings of most existing studies indicated that those with better social support were expected to have a lower risk of depression and less severe depression/depressive symptoms [46,47,48,49,50,51,52]. Nevertheless, most of these studies do not particularly target stroke patients, who have the health conditions that are related to depression [46,47,48,49,50,51,52]. Given the distinct challenges that arise from stroke—long rehabilitation process, difficulties to perform daily activities, loneliness, cost of medical needs and so forth–social support and depression might be associated differently among stroke patients than other populations or might be associated [15,53]. Moreover, most review articles and meta-analyses on relevant topics either broadly describe psycho-social factors or are more focused on intervention programs in clinical settings, which were fundamentally different from social support that is perceived in daily life [54,55,56,57]. Summary and evaluation of studies are crucial to understand what is known regarding social support and depression among stroke patients.

This literature review aims to evaluate the current research studying the association between social support and depression, specifically among stroke patients.

## 2. Materials and Methods

This literature review includes peer-reviewed journal articles that examined the association between social support and depression among stroke patients. PubMed was used to search for the terms “depression and ‘social support’ and stroke”. “Social support” was quoted in the search box to exclude irrelevant results. The most recent search date was 12 March 2023, and this review includes papers published up to that date. The year range was restricted to between 2016 and 2023 (including 2016 and 2023) so that only recent studies were included. According to the History and Search Details of Pubmed, the search terms were automatically mapped to MeSH terms, which were translated as following: depression: “depressed”[All Fields] OR “depression”[MeSH Terms] OR “depression”[All Fields] OR “depressions”[All Fields] OR “depression’s”[All Fields] OR “depressive disorder”[MeSH Terms] OR (“depressive”[All Fields] AND “disorder”[All Fields]) OR “depressive disorder”[All Fields] OR “depressivity”[All Fields] OR “depressive”[All Fields] OR “depressively”[All Fields] OR “depressiveness”[All Fields] OR “depressives”[All Fields]; stroke: “stroke”[MeSH Terms] OR “stroke”[All Fields] OR “strokes”[All Fields] OR “stroke’s”[All Fields]. More specific terms such as “hemorrhagic stroke”, “ischemic stroke”, or “ischemia attack” were unnecessary because the Mesh Term “stroke” already covered all the specific situations below “stroke” in the MeSH term tree structure. Thus, although only the above search terms were used, the search results included all the studies that might meet the inclusion criteria for this literature review in the PubMed database. This search resulted in 172 articles, of which one article was retracted. Figure 1 shows the steps of searching and selection of articles.

Abstracts of the remaining 172 articles were read for selection based on inclusion and exclusion rules. For a study to be included, it must be an original observational study, for the exposure of interest was social support that can only be perceived from daily life. Cohort studies, cross-sectional studies, and case-control studies were acceptable. The exposure and the outcome of interest must be specifically defined with explicit operationalization. Social support/perceived social support must be the primary exposure or one of the primary exposures, and depression/depressive symptoms/depressiveness must be the primary outcome or one of the primary outcomes. The target population had to be stroke patients. There was no restriction on specific types of stroke—ischemic stroke, hemorrhagic stroke, and transient ischemic attack (TIA) [58]. Measures of association must be provided for a study to be included, regardless of the statistical significance level.

Among the 172 articles found by search terms, 11 studies were literature reviews or systematic reviews; four studies were meta-analyses; 142 studies were identified as irrelevant ones that did not study the target association—social support and depression, or targeted other populations than stroke patients. Besides the fact that the article types did not meet our inclusion criteria, the literature reviews and meta-analyses did not study the association of interest among stroke patients although these 15 articles showed up in the search results. One experimental study designed and implemented social support interventions and compared depression levels between the intervention group and the control group [59]. Excluding experimental studies had a conceptual reason: the exposure differed from the one that the topic review focuses on—the social support status perceived by subjects in their daily life instead of a social support intervention. After all criteria were applied, eight studies were left for this topical review, including five prospective cohort studies, and three cross-sectional studies [60,61,62,63,64,65,66,67].

APA PsycINFO was used as a complementary database. The same search strategies and inclusion and exclusion criteria were used to find if there were additional appropriate papers. The same search terms resulted in 83 papers (the last search date: 12 March 2023), including two prospective cohort studies that met all the criteria [62,66]. One of the cohort studies overlapped with the selected papers from PubMed [66]. The other study was selected [62]. Thus, nine studies were selected for the final review [60,61,62,63,64,65,66,67,68].

## 3. Results

### 3.1. Samples and Populations

The main characteristics of the selected studies are listed in Table 1. The studies included five prospective cohort studies and three cross-sectional studies. Additionally, the study by Ladwig et al. analyzed data from two cohort studies [68]. All the studies except Ladwig et al. enrolled adults (age ≥ 18 years) that had been diagnosed with stroke [60,61,62,63,64,65,66,67,68]. The average age of eight studies ranged from 56.4 to 70.8 years [61,62,63,64,65,66,67,68]. The gender distribution was roughly balanced in six studies, where neither gender exceeded 60% [61,63,65,66,67,68]. The most unbalanced gender distribution (29.1% females) appeared in the study by Wei et al. [62]. One article did not reveal mean age or gender distribution [60]. The sample size of respondents ranged from 33 in a prospective cohort study to 710 in a cross-sectional study [60,61,62,63,64,65,66,67]. The mean sample size was 230.0, the median 160.5, and the standard deviation 224.0. Eight studies focused on inpatients from hospitals, rehabilitation centers, and/or clinics [61,62,63,64,65,66,67,68]. One study recruited stroke patients, of which the source was not specified in the article [60].

The population of interest in this review was stroke patients, regardless of stroke types and lengths of time between the occurrence of stroke and data collection or enrollment. The cross-sectional study by Olibamoyo et al. specified that stroke must have occurred in the six months prior to enrollment [56]. Six studies had restrictions on specific types or the status of stroke that participants had [60,61,62,66,67,68]. The cohort study by Volz et al. restricted the participants to only patients with a first ischemic stroke diagnosed over four weeks prior to enrollment [66]. Schöttke et al. only enrolled patients with either acute cerebral infarction or intracerebral hemorrhage over 24 h prior [67]. The cross-sectional study by Ahmed et al. included only patients with a history of acute ischemic stroke confirmed with neuroimage evidence [61]. Villain et al. limited their sample to adult patients with first ischemic stroke diagnosed by a neurologist with clinical evidence and magnetic resonance image (MRI) [60]. The two samples in the study by Ladwig et al. only included ischemic stroke patients; clinical evidence was unspecified in the paper [59]. Wei et al. excluded those with hemorrhagic stroke [62]. The other two studies did not add restrictions on a time duration from the stroke occurrence or stroke types [63,64].

### 3.2. Social Support and Depression Measurements

The study by Olibamoyo et al. measured social support using only one question in a demographic questionnaire [65]. The eight other studies measured social support with validated psychological scales [60,61,62,63,64,67,68]. The scales used to assess social support included the Multidimensional Scale of Perceived Social Support (MSPSS), the Perceived Social Support Questionnaire (F-SozU, both the 22-item version F-SozU K-22 and the 14-item version F-SozU K-14), Social Support Rating Scale (SSRS), and the MOS Social Support Survey (MOS-SSS) [69,70,71,72,73,74]. For depression, the outcome of interest, one study confirmed subjects’ depression using diagnostic evaluations at three time points [75]. One study diagnosed depression based on Diagnostic and Statistical Manual of Mental Disorders (DSM-IV) [76]. Seven studies assessed depression using other scales, which included the Mini-International Neuropsychiatric Interview (MINI), the Hospital Anxiety and Depression Scale (HADS), Cornell Scale for Depression (CDS), the Hamilton Depression Rating Scale (HDRS-17), the Center for Epidemiologic Studies Depression Scale (CES-D), the Patient Health Questionnaire-9 (PHQ-9), and Geriatric Depression Scale (GDS) [77,78,79,80,81,82,83].

### 3.3. Main Findings

Despite the differences in sample sizes, study design, and analytical methods that existed across the studies, there was one consistent facet in the findings: eight studies found negative associations between social support and depression among stroke patients [60,61,62,63,64,65,67,68]. The association estimates in eight of the nine studies indicated that stroke patients with poorer social support were more likely to suffer from depression or were expected to have worse depression status [60,61,62,63,64,65,67,68]. Due to the different parameterizations of social support and depression, the estimates could not be compared directly between studies on ischemic-stroke patients and those on patients with any type of stroke. Nevertheless, both types of studies reported odds ratios (ORs) greater than 1.3 (absolute value) [61,67].

As the only selected study that measured social support with one single item, the study by Olibamoyo et al. categorized social support into “poor”, “fair”, and “good” [56]. They found the adjusted OR that was the furthest from the null: OR (fair vs. good) = 7.6, 95% CI: (0.96, 59.614), OR (poor vs. good) = 92.4, 95% CI: (3.71, 2296.83) among patients with any type of stroke [65]. The article did not address the possible reasons for the extreme estimates. To measure social support as a construct, using only one item is likely to yield low reliability. The OR of categorized social support is not comparable with the ORs in other studies, where social support was represented by a scale score in regression models. Besides this, the extremely wide confidence intervals were not informative.

In the studies that measured social support with psychological scales and applied logistic regression models, the adjusted OR furthest from the null was in the study by Ahmed et al.: 0.66, 95% CI: (0.47, 0.94) for a one-unit increase in the MSPSS scale, the OR closest to the null in the study by Volz et al.: 0.95 (*p* = 0.03) for a one-unit increase in the F-SozU scale [61,62,66,67]. Ahmed et al. recruited 68 ischemic-stroke patients and adjusted for the fewest covariates in the models: stroke severity at admission, poststroke disability after 3 months, and discontinuation of rehabilitation. No demographics were included in the models, while five other studies included several, although not identical, demographic covariates [60,61,63,64,65,67]. The study by Wei et al. had a larger sample size (368 adult patients with any type of stroke except hemorrhagic stroke) than the study by Ahmed et al., and did not adjust for demographic covariates [62]. The association between social support at baseline and depression three months afterwards (adjusted OR = 0.57, *p* = 0.001) was found close to that in the study by Ahmed et al. [61,62].

Four studies measured social support with psychological scales and fitted linear regression models [60,63,64,68]. Among the four studies, both the study by Wang et al. and the study by Babkair et al. were cross-sectional studies and recruited stroke patients through convenient sampling [63,64]. The sample sizes were 800 and 135 adult stroke patients, respectively [63,64]. The estimated coefficients in linear models were β = −0.111 (*p* < 0.01) and β = −0.31 (*p* < 0.001) [63,64], respectively, meaning that a single-unit increase in the scores measured by their selected social support scales was respectively associated with decreases of 0.111 and 0.31 in the depression scales. Both studies adjusted for demographics and variables measuring the subjects’ physical health at baseline [63,64]. The main difference in the two studies that might lead to the difference in effect estimates was the measurement instruments used; Wang et al. used the PSSS scale to measure social support, and assessed depression with CES-D while Babkair et al. used MOS-SSS for social support and PHQ-9 for depression [63,64,69,72,81,82]. Villain et al. broke down perceived social support at admission, the exposure, into multiple components: medical attention from medical staff, moral support from medical staff, moral support from family and friends, and material support from family and friends [60]. The ischemic stroke patients had these four components assessed through ecological momentary assessment (EMA) [60,84]. The estimated coefficients in the models adjusting for age, sex, anxiety, and depression levels at hospital admission were medical attention from medical staff: γ = −0.019 (SE = 0.055, t ratio = −0.357), moral support from medical staff: γ = −0.051 (SE = 0.043, t ratio = −1.191), moral support from family and friends: γ = −0.097 (SE = 0.046, t ratio = −2.141), and material support from family and friends: γ = 0.008 (SE = 0.055, t ratio = 0.161) adjusting for age, sex, and baseline depression levels [60]. All the four components of social support were found to be negatively related with depression three months after hospital discharge in the cohort study. The effect of moral support from family and friends was the strongest and statistically significant. The sample size of this study was 33 after dropping those lost to follow-up, the smallest among the eight studies. Both cohort studies in the article of Ladwig et al. recruited only ischemic stroke patients [68]. They treated social support scores (one by F-SozU K-22, and one by F-SozU K-14) as continuous variables [68]. Depression was also a continuous variable in both studies; one used the GDS score and one used the PHQ-9 score [68]. Consistent with Wang et al., Babkair et al., and Villain et al., Ladwig et al. found that social support was negatively associated with the risk of poststroke depression six months later (Study A: social support measured by F-SozU K22, depression measured by GDS-15, β = −1.91, 95% CI: (−2.71, −1.11), Study B: social support measured by F-SozU K14, depression measured by PHQ-9, β = −2.69, 95% CI: (−3.92, −1.47)) [68]. Besides the association of different social support levels between individuals and depression, Ladwig et al. also found a protective effect of within-person change in social support six months after the acute phase on depression (β = −0.14, 95% CI: (−0.22, −0.05)) [68].

Eight of the selected studies found statistically significant associations, but one did not. The prospective cohort study conducted by Schöttke et al. had a sample of 78 patients with any type of stroke after a 3-year follow-up [67]. The study looked into the association between perceived social support at baseline and depression three years after stroke onset [67]. The adjusted covariates in the study included sex, age, functional impairment, depression at the acute phase of stroke, and estimated the association with (adjusted OR = 1.825 (95% CI 0.593, 5.617)) and without the interaction (adjusted OR = 1.355 (95% CI 0.516, 3.560), *p* = 0.537) between functional impairment and perceived social support at the acute phase [67]. The point estimates of OR were greater than 1, which conflicted with the findings in all the other seven studies. This study did not adjust for stroke severity, which was an important covariate in multiple other studies [61,65,66]. Besides, Schöttke et al. measured depression both at enrollment and three years afterwards at follow up (the longest duration between the baseline when social support was measured and the measurement of depression) [67]. The final model predicting depression three years after enrollment included the depression level at baseline as a predictor, which was unique in all the selected studies except for the study by Villain et al. [60,61,62,63,64,65,66,67,68].

Six studies adjusted for demographics, among which five studies adjusted for age, and four adjusted for sex review [60,63,64,65,67,68]. Other demographic covariates were not shared by most studies. Besides demographics, variables related to physical health conditions, particularly stroke severity and physical disability, were important covariates as seven studies included at least one of the two variables in the final model [61,62,63,64,65,66,67,68].

## 4. Discussion

The primary purpose of this literature review was to examine recent literature on social support and depression among stroke patients. PSD is a common condition that leads to suffering of patients and is a key problem to address in stroke treatment. Eight studies consistently found that depression was negatively associated with social support and the association was statistically significant. In total, four studies recruited stroke patients without restriction on the stroke types; four were based on only ischemic-stroke patients and one excluded hemorrhagic stroke patients. Another difference across the eight studies was the restriction on time of measuring the variables. Of the eight studies with significant results, all cohort studies took measurements at baseline (hospital admission), but follow-up time varied from 90 days to 6.1 months; one cross-sectional study was restricted to patients with a stroke history at least 6 months prior [61,65,66]. There was no substantive distinction in the findings on different stroke patients in terms of what type of stroke and when PSD and social support were measured relative to stroke onset. Social support appeared consistently associated with lower depression severity or risk of having depression. This is also consistent with the association of social support and depression among other populations and under other scenarios, such as individuals at the time of social isolation and social distancing due to the COVID-19 pandemic [85], and among elderly people [86,87,88].

The only study that had a discrepancy with the others regarding the target association was that by Schöttke et al., which did not restrict stroke types [67]. The prospective cohort study by Schöttke et al. did not find a statistically significant association and the direction of the association conflicted with the other seven studies [67]. This discrepancy might be caused by a variety of factors: the covariates selected biasing the estimate, selection bias caused by a high proportion of loss to follow-up, and uncontrolled confounding especially given the long interval between the baseline and the follow-up three years after stroke. The other eight studies observed statistically significant associations within a shorter period after stroke onset [60,61,62,63,64,65,66,68]. Assuming that the findings of Schöttke et al. are unbiased, the discrepancy might indicate that social support plays a weaker role in the later phase of stroke [67]. Schöttke et al. recruited patients from three rehabilitation clinics; the target population for inference was not exactly the same as the other studies in this review [67]. The inclusion criteria also varied across studies. For example, the study by Schöttke et al. restricted the participants to stroke patients with acute cerebral infarction and those with intracerebral hemorrhage [67]. In addition, the study required that the documentation of neurological symptoms must have exceeded 24 h prior to the study enrollment [67]. The other studies used different inclusion and exclusion criteria, which yielded different source populations in terms of stroke conditions. Another potential reason for the discrepancy might be the varying measures for social support and depression, which lead to an undeniable fact that the selected studies, in essence, evaluated associations slightly different from each other even if regardless of the covariates and populations.

Based on the comparison of the studies, including demographics did not systematically improve or reduce effect estimates of social support [60,61,62,63,64,65,66,67,68]. Neither the study with the OR furthest from the null nor the study with the OR closest to the null included adjustment for demographics [61,66]. The main difference in estimated effect sizes appears to be more affected by sample size and measuring instruments for social support and depression. The study by Ahmed et al. found the largest effect size (as OR), which measured social support with MSPSS and assessed depression with HADS [52,60,69]. The study by Volz et al. reported the smallest effect size (as OR), where social support was measured with HDRS-17 and depression with DSM-IV [66,76,89]. The nine studies were not enough to present patterns such as which combination of the scales would yield stronger or weaker effect estimates of social support on depression.

Five of the selected studies were prospective cohort studies, three were cross-sectional studies, and one was a secondary analysis based on longitudinal data [60,61,62,63,64,65,66,67,68]. All the cohort studies investigated the association between social support at baseline and depression after a follow-up period. The secondary analysis examined the effect of change in social support on depression after follow-up. Thus, temporality was clear in the data analysis of the cohort studies as well as the secondary analysis, which is an essential prerequisite for causality. Given the consistent findings, it is plausible to conclude that social support has a protective effect against PSD, which is itself a serious mental problem and is associated with poor treatment adherence and increased probabilities of bad health outcomes including stroke recurrence and death.

### 4.1. Limitations

Several limitations must be carefully considered regarding the selected studies. First, only one study explicitly explained that it used a probability sampling method, which was good for the representativeness of the sample. The other studies either used convenient sampling or did not introduce the sampling method. This weakness rendered a potential concern to the external validity of the study findings. Second, some of the studies had moderate sample sizes, the smallest of which was 44 enrolled and 33 followed up [60]. It is plausible to worry about random errors that might have biased the results. Third, none of the selected studies conducted sensitivity analysis [60,61,62,63,64,65,66,67,68]. Unlike randomized controlled trials, an essential concern in observational studies is always uncontrolled confounding that may explain away the association [90]. Thus, it is crucial to conduct sensitivity analysis in observational studies, which is lacking in all the selected studies. The robustness of the study findings to uncontrolled confounding remains unknown. This limitation also makes it impossible to identify which study built the most efficient model that estimated the association with the least necessary covariates while adjusting for confounders adequately. An essential piece of evidence, though not a necessary condition, is missing for drawing causal conclusions due to this fact. Fourth, the three cross-sectional studies could not provide temporality of social support and depression. Thus, the association estimates provided by these three studies, either adjusted or unadjusted, could not infer causality.

Other limitations included but were not limited to: (1) One study had a 45% dropout rate, potentially impairing the representativeness and introducing bias, the direction of which could be either away from the null or towards the null [67]; (2) patients with severe functional impairment or extremely poor cognitive status were excluded from some studies, which also limited the generalizability and transportability and could bias the results if the targeted population was not adjusted to those that did not include the severe patients in interpretation. Existing studies have found worse stroke severity to predict more severe depression [17,91]. Despite this, no evidence was found on the association between stroke severity at baseline (at onset or at hospital admission) and social support. The direction of bias caused by exclusion remains unknown; (3) Hospital setting also might have lowered the generalizability of study findings, for more of the stroke patients were still at the acute phase at the time of study or more severe than patients at home [92].

### 4.2. Strengths

The selected studies had noticeable strengths despite the limitations listed above: (1) Temporality of social support and depression, a necessary but insufficient condition for causality, was clear in the five prospective cohort studies; (2) The relatively large sample sizes of several studies rendered their effect estimates less prone to random error than those with small sample sizes. The largest sample size was 710 in the study by Wang et al. where the linear coefficient of social support score was −0.111 (*p* < 0.01), adding to the power of the study and the strength of our conclusion on social support and depression [63]; (3) Although covariates adjusted for varied across the studies, eight of the studies reported statistically significant associations and six of them were strong associations. This was evidence in favor of causality despite the absence of sensitivity analysis. (4) Different measuring instruments for social support and depression yielded the same direction of the targeted association, the only difference of which was the magnitude of the effect estimates. This fact further supports the robustness of the observed protective effect of social support against depression among stroke patients.

### 4.3. Future Research Directions

Several advancements should be made in future studies. First, add sensitivity analysis to observational studies on the association between social support and depression among stroke patients. This is due to the nature of observational studies, of which the purpose is to measure the robustness of effect estimates to uncontrolled confounding and provide evidence of causality [90,93,94]. The research question of this literature review by itself eliminates the possibility of experimental study designs because the exposure of interest is social support perceived by subjects in their life. In an experimental study, the exposure would become a support intervention, which is different from the exposure of interest. One experimental study was excluded in article selection because of this reason. Second, more combinations of covariates both as potential confounders and potential effect modifiers should be tested out in regression analysis. Variables measuring or reflecting the physical health conditions of stroke patients could make good covariates, such as NIHSS score and Barthel Index that measure stroke severity and activities of daily life. These two variables were included as covariates (confounders) in multiple selected studies, but not all of them. It has not yet been tested as to whether these two variables are effect modifiers. Using these variables as a base, researchers can try adding other covariates to regression models. Third, it might be more informative if future studies explicitly explain the sampling method (i.e., whether probability sampling or nonprobability sampling such as convenient sampling). Fourth, in order to increase sample sizes and avoid a high dropout rate, certain solutions need to be taken such as increasing the number of hospitals to recruit patients from, adding incentives for enrollment and responses during the follow-up period, and shortening the observation period if feasible and not conflicting with research questions. Lastly, researchers are encouraged to implement qualitative studies, cross-sectional studies, or even short-term cohort studies if feasible given the resources, to explore psycho-social factors that affect the association between social support and depression, such as prior diagnosis of depression or other mental conditions before the stroke onset.

## 5. Conclusions

Overall, social support appears to be negatively associated with depression among stroke patients. Stroke patients with better social support conditions are expected to have a lower risk of depression or less severe depression. The association is statistically significant and supported by most recent studies. Multiple methodological improvements can still be made to accumulate valid evidence of causality on this association.

From the perspective of clinical practice, this review examined the quantitative importance of social support, particularly the support perceived by stroke patients, in lowering the risk and severity of post-stroke depression, which then may improve treatment adherence and reduce poor health outcomes including stroke recurrences and deaths. Care providers are encouraged to evaluate the depression/depressive level and social support of stroke patients as a routine and help address factors that hinder the reception and perception of social support.

## Figures and Tables

**Figure 1 ijerph-20-07157-f001:**
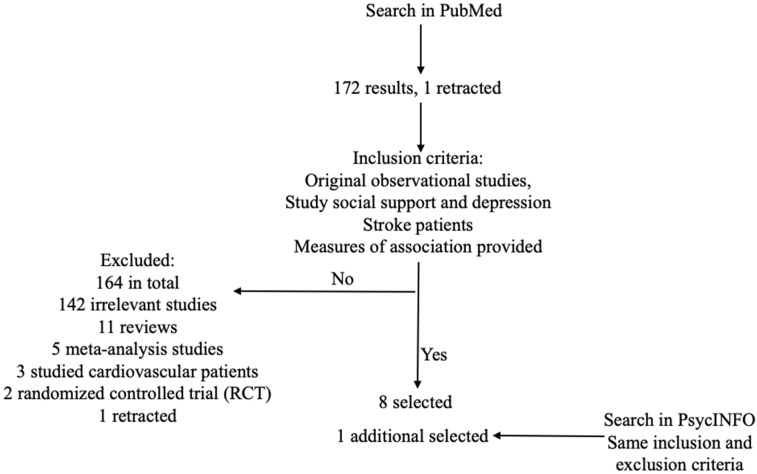
Search and selection of articles.

**Table 1 ijerph-20-07157-t001:** Characteristics of Selected Studies.

Citation	Country and Setting	Study Type (Year) and Sampling Method	Sample Description	Inclusion Criteria	Exclusion Criteria	Tools Measuring Exposure Variable	Tools Measuring Outcome Variable	Statistics	Main Findings
Olibamoyo et al., 2019 [65]	Nigeria, inpatients, a university teaching hospital	A cross-sectional study Simple random sampling.	112 adults, age: 18–65	Clinical history of stroke not less than 6 months prior, confirmed by neuroimaging.	History of any psychotic disorder, history or pre-stroke mood disorder, language impairment severe enough to prevent neuro-psychiatry assessment, history of other central nervous disorders and acute illness preventing proper assessment	Perceived social support, assessed by one question in a demographic questionnaire, categories: good, fair, and poor	Depression, assessed by MINI English Version 5.0	Poor social support vs. good social support adjusted OR: 92.4 (3.71–22,296.83), *p* = 0.006	Perceived poor social support, higher number of previous admissions of stroke, younger age, and unemployment are associated with increased risk of depression.
Ahmed et al., 2020 [61]	Saudi Arabia, Inpatients, two tertiary stroke centers	A prospective cohort study (90-day follow-up). Convenient sampling.	68 adults, mean age: 56.72, age range: 29–75	Acute ischemic stroke, confirmed clinical and radiological evidence/	Fever, cough, or any acute respiratory symptoms, any chronic psychiatric or neurocognitive disorder, pre-existing chronic neurological disorders, chronic debilitating medical disorders	Perceived social support assessed by (MSPSS).	Post-stroke depression (PSD) and post-stroke (PSA) assessed by HADS	MSPSS score unadjusted OR: 0.44 (0.26–0.75), *p* = 0.003; adjusted OR: 0.66 (0.47–0.94), *p* = 0.002	Lower social support score, worse stroke severity at admission, and worse disability status are associated with an increased risk of post-stroke depression.
Schöttke et al., 2020 [67]	German, inpatients, three rehabilitation clinics	A prospective cohort study (3-year follow-up). Sampling method not introduced.	174 patients, mean age: 67.51	Either acute cerebral infarction or intracerebral hemorrhage, neurological symptoms over 24 h prior.	The need for intensive medical treatment, artificial respiration, intensive treatment of body injuries and heightened intracranial pressure, and severe aphasia.	Perceived social support assessed by F-SozU at acute phase	PSD assessed by structured clinical interview and Functional impairment assessed by BI	Without the interaction term, F-SozU adjusted OR: 1.355 (0.516, 3.560), *p* = 0.537. With the interaction term, F-SozU adjusted OR: 1.825 (0.593, 5.617), *p* = 0.295.	Perceived social support is not found to have a statistically significant association with poststroke depression 3 years from stroke, adjusting for age, sex, functional impairment, and depression at acute phase with or without the interaction between functional impairment and social support at acute phase.
Villain et al., 2017 [60]	France, stroke patients, setting not specified	A prospective cohort study (3-month follow-up). Convenient sampling.	44 adults enrolled, 33 followed up	First mild ischemic stroke diagnosed by a neurologist, confirmed with clinical evidence and MRI, NIHSS score ≤ 6.	A history of dementia (Mini-Mental State Examination < 24), severe aphasia, visual or motor handicap, or a history of major depression prior to stroke.	Perceived social support assessed by Ecological Momentary Assessments immediately after stroke	Depression assessed by HDRS-17 at follow-up	Medical attention γ: −0.019, SE: 0.055, t ratio: −0.357; Moral support (medical staff) γ: −0.051, SE: 0.043, t ratio: −1.191; Moral support (family, friends) γ: −0.097, SE: 0.046, t ratio: −2.141; Material support (family, friends) γ: 0.008, SE: 0.055, t ratio: 0.161	Lower perceived moral support from family and friends are associated with an increased risk of depression, conditional on medical attention, adjusting for age, sex, anxiety, and depression levels at hospital admission.
Volz et al., 2016 [66]	German, inpatients, two inpatient rehabilitation centers	A prospective cohort study (6.1-month follow-up) Sampling methods not introduced.	88 adults, mean age: 66.35, age SD: 10.70, age range: 44–90	First ischemic stroke ≥ 4 weeks prior, education ≥ 8 years, comprehension Token test > 12, age over 40, less than 12 weeks deviation from follow-up interval		social support assessed by F-SozU	Depression diagnosed by DSM-IV	F-SozU adjusted OR = 0.95, *p* = 0.03	Lower social support is associated with increases in later depression, adjusting for physical disability status, pre-stroke mental illness, stroke severity, cognitive status, and general self-efficacy and early depressiveness.
Wang et al., 2019 [63]	China, inpatients from community hospitals in five major cities in a province	A cross-sectional study. Convenient sampling.	800 stroke patients enrolled, 710 responded, age: 30–90	Stroke patients, able to read.	Missing items in questionnaires	Perceived social support assessed by the Chinese version of MSPSS	Depressive symptoms assessed by CES-D	MSPSS adjusted β: −0.111, *p* < 0.01 in linear regression	Lower social support is associated with increases depressive symptoms, adjusting for age, gender, marital status, education, residence types, chronic disease, monthly income, medical payment types, activities of daily living, hope, resilience, and self-efficacy.
Babkair et al., 2021 [64]	Saudi Arabia, inpatients from three hospitals	A cross-sectional study Convenient sampling.	135 adults, history of stroke, age ≥ 18	Stroke patients, age ≥ 18 years, able to comprehend and communicate in Arabic	Conditions that limit ability to complete a survey (cognitive impairment, dementia, aphasia, and chronic psychiatric diagnoses except previous depression)	Social support assessed by MOS-SSS	Poststroke depressive symptoms assessed by PHQ-9	MOS-SSS adjusted β: −0.31, *p* < 0.001	Lower perceived social support is associated with increased depressive symptom level, adjusting for sex, marital status, employment, income, perceived stress, and functional independence.
Wei et al., 2016 [62]	China, inpatients from a university hospital	A prospective cohort study (3-month follow-up)	368 adult stroke patients, overall age statistics not provided	Diagnosis of acute stroke confirmed based on CT or MRI findings within 7 days after stroke onset	Hemorrhagic stroke, unusual causes such as dissections, venous infarction or moyamoya disease, transient ischemic attack (TIA) without progression to stroke, communication problems, severe neurologic or medical conditions, score ≤ 23 on the Mini-Mental State Examination (MMSE)	Social support assessed by SSRS	Poststroke depression assessed by BDI	The degree of social utilization section of SSRS adjusted β: −0.558, *p* = 0.001. Converted to adjusted odds ratio: 0.57	Higher degree of social utilization, a dimension of social support, is associated with decreased poststroke depression, adjusting for sensory dysfunction and motor dysfunction at admission, and coping strategies including avoidance level and acceptance vs. resignation.
Ladwig et al., 2023 [68]	Study A: German, inpatients from two rehabilitation clinics; Study B: German, inpatients from the stroke unit of an acute hospital	Secondary data analysis based on two cohort studies (denoted studies A and B, 6 month-follow-up in) Sampling methods not introduced.	273 (Study A) and 226 (Study B). Study A mean age: 63.9, age SD: 10.9, age range: 42–92. Study B mean age: 64.2, age SD: 10.2, age range: 42–90.	Both studies: ischemic stroke patients with sufficient language comprehension. Study B further required sufficient cognition.	Both studies: terminal or impairing disease other than stroke.	Social support assessed by F-SozU K22 (Study A) and F-SozU K14 (Study B);	Depressive symptoms assessed by GDS_15 (Study A) and PHQ-9 (Study B)	Study A: F-SozU adjusted β = −1.91, 95% CI: (−2.71, −1.11). Study B: F-SozU K-14 adjusted β = −2.69, 95% CI: (−3.92, −1.47). Difference of social support adjusted β = −0.14 (−0.22, 0.05).	Social support is negatively associated with the risk of post-stroke depression, adjusting for history of mental disorder, stroke severity, physical. The change of social support (follow-up minus baseline) is also a protective factor against depression.

Note. NIHSS = National Institute of Health Stroke Scale; MNSE = Mini Mental State Examination; MINI = Mini International Neuropsychiatric Interview; mRS = Modified Ranking Scale; MSPSS = Multidimensional Scale of Perceived Social Support; HADS = Hospital Anxiety and Depression Scale; HDRS-17 = Hamilton Depression Rating Scale; HARS = Hamilton Anxiety Rating Scale; BI = Barthel Index; GSE = General Self Efficacy Scale; MMST = Mini-Mental-State-Test; F-SozU = the Perceived Social Support Questionnaire (the 22-item version); F-SozU K-14: the 14-item version of F-SozU; CDS = Cornell Scale for Depression; BDS = Geriatric Depression Scale; EPQ = Eysenck Personality Questionnaire; SSRS = Social Support Rating Scale; LES = Life Event Scale; TAS = 20 items Toronto Alexithymia Scale; CES-D = Center for Epidemiologic Studies Depression Scale; AHS = Adult Hope Scale; RS-14 = Wagnild and Young Resilience Scale-14; PSS-10 = Perceived Stress Scale-10: FIM = Functional Independence Measure; MOS-SSS = Medical Outcomes Study Social Support Survey; PHQ-9 = Patient Health Questionnaire-9; GDS: Geriatric Depression Scale; GDS-15: a shorter version of GDS; —Discussion Importance. Note. For all the studies that built multiple regression models, only the adjusted statistics of the final model are reported in the table.

## Data Availability

Not applicable.

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
