# Peer review of "Social Support and Depression among Stroke Patients: A Topical Review"

_ijerph, 2023, doi:10.3390/ijerph20247157_

Round 1

Reviewer 1 Report

Comments and Suggestions for Authors

Dear authors,

congratulations on the presented paper, I applaud your efforts to summarize the findings of different studies on stroke patients and their mental health. You have mostly provided organized and comprehensive comparisons of presented findings, elaborated on any discrepancies, and proposed future research directions.

However, there is still room for improvement. The introduction part regarding depression is not informative enough and is based mostly on the issue of underreporting and somatic symptoms. It should be rewritten and linked to patients overall quality of life. The part about stroke could be rearranged. 

We are not supposed to comment on insignificant results, no matter the difference or the effect raw data are showing; so please rephrase those sections into ''not significant'' and do not comment on the +/- sign.

I would add another 1-2 keywords for better citing opportunities, perhaps aimed at the chosen methodology or samples.

Emphasize the PRISMA model guidelines and steps and cite them appropriately. 

Please elaborate a bit further on your inclusion/exclusion criteria. 

I have suggested you add some additional data on 3. 1. section (Sample)

Please avoid anthropomorphisms.

The first paragraph of the Conclusion section should have been elaborated in the discussion section, and in greater length since it is the aim of this study - to provide an answer as to weather social supports can be linked to lower depression levels.

I am still a bit unsure whether your paper represents a significant contribution to the field; and thus missing the valid implication for this paper. I suggest you add a few recent references on this matter and what incited you to choose this subject in the first place. What problem do your results help to solve or understand? 

And just a curiosity - why isn't Rick Lezenby listed as one of the authors?

Reviewer 2 Report

Comments and Suggestions for Authors

Thank you for the opportunity to review your literature review of PSD in patients with stroke. I want to first say that I found your article very easy to read and follow, so well done. The intro was overall adequate as was the review of the 9 articles selected for your review. However, I do have a few comments below that I think are worth considering for publication.

First, in the intro, I think you did a good job with reviewing the literature. What was missing to me is why this review was important to be done. You established that there was already an existing link between social support and PSD in your review of the literature. So, what makes your study valuable in consideration of these variables? I think the more that you can link what maybe was lacking prior to 2016 or in other reviews on this topic would be a lot more helpful. In your literature search, you indicated you found 11 review articles and 5 meta-analysis studies between 2016-2023. What were their findings, and how did your study differ from them?

Second, I agree that there is no way to do an experimental study to test the connection between PSD and social support. It would also be very time-consuming to do a longitudinal study to examine all these factors. I do wonder, though, if a suggestion of at least qualitative research in this area could be helpful to examine some of these variables that could be affected by pre-stroke status, such as any prior history of depression and level of social support pre-stroke.

Finally, just some grammatical stuff to review, mostly related to commas. Examples are:

Line 59:  "...among other factors, including concerns of being..."

Line 60:  "...and being referred to a psychiatrist, often hinders..."

Line 92:  "...was Mar. 12, 2023, and this review..."

Line 202:  Delete extra space between "Schottke  and colleagues"

Comments on the Quality of English Language

See above comments.

Otherwise, it was very readable and flowed nicely.

Reviewer 3 Report

Comments and Suggestions for Authors

Dear Author

The paper is characterized by originality, and it is well structured.  I consider that the abstract gathers substantial elements to catch the reader's interest, my recommendation would be to expand a little on the results.

Regarding the introduction and methodology, I consider that the relationships between the variables and the author's interests are described, and the search methodology is correct and up to date.

 However, I consider that since this is a peer-reviewed article that generalizes results, it is convenient to broaden the context of the discussion that represents the central element of its analysis.

Round 2

Reviewer 1 Report

Comments and Suggestions for Authors

I find the paper significantly improved as authors followed all the instructions they received.